# Multi-focused image fusion algorithm based on multi-scale hybrid attention residual network

**Tingting Liu**[1,2]*, **Mingju Chen**[1,2], **Zhengxu Duan**[1,2], **Anle Cui**[1,2]

**1** Sichuan Key Laboratory of Artificial Intelligence, Sichuan University of Science and Engineering, Yibin, Sichuan, China, **2** School of Automation and Information, Sichuan University of Science and Engineering, Yibin, Sichuan, China

* 321081104112@suse.edu.cn

**Data Availability Statement:** All relevant data are within the paper.

**Funding:** This research was funded by Natural Science Foundation of Sichuan, China

## Abstract

In order to improve the detection performance of image fusion in focus areas and realize end-to-end decision diagram optimization, we design a multi-focus image fusion network based on deep learning. The network is trained using unsupervised learning and a multi-scale hybrid attention residual network model is introduced to enable solving for features at different levels of the image. In the training stage, multi-scale features are extracted from two source images with different focal points using hybrid multi-scale residual blocks (MSRB), and the up-down projection module (UDP) is introduced to obtain multi-scale edge information, then the extracted features are operated to obtain deeper image features. These blocks can effectively utilize multi-scale feature information without increasing the number of parameters. The deep features of the image are extracted in its test phase, input to the spatial frequency domain to calculate and measure the activity level and obtain the initial decision map, and use post-processing techniques to eliminate the edge errors. Finally, the decision map is generated and optimized, and the final fused image is obtained by combining the optimized decision map with the source image. The comparative experiments show that our proposed model achieves better fusion performance in subjective evaluation, and the quality of the obtained fused images is more robust with richer details. The objective evaluation metrics work better and the image fusion quality is higher.

## Introduction

Due to various limitations of hardware devices, the depth of field of optical lenses is limited [1]. Images captured by a single sensor or a single camera cannot effectively and comprehensively describe the imaging scene. Objects within the depth of field are clear, while other scene contents may be blurred. This makes it difficult to capture an image in which all the objects present are in focus [2]. To extend the depth of field of the lens and improve the quality of the image, an intelligent and efficient technique is needed. This technique should integrate multiple images with different focus planes so that all objects are in full focus. A method to achieve this goal is known as a multi-focus image fusion algorithm.

(2023NSFSC1987,2022ZHCG0035); The Key Laboratory of Internet Information Retrieval of Hainan Province Research Found(2022KY03); the Opening Project of International Joint Research Center for Robotics and Intelligence System of Sichuan Province(JQZN2022-005); Sichuan University of Science & Engineering Postgraduate Innovation Fund Project, grant number Y2022130. The funders had no role in study design, data collection and analysis, decision to publish, or preparation of the manuscript.

**Competing interests:** The authors declare no competing interests.

A multi-focus image fusion algorithm can combine meaningful information from multiple source images to obtain a single fused image with more detailed information and enhanced features than a conventional bokeh image. Compared to the original image, the multi-focus image fusion algorithm can obtain an image with a more accurate description of the scene details, which is more beneficial for subsequent applications. Multi-focus image fusion algorithms have been applied in many fields, such as micro-scopic image fusion [3], medical diagnosis [4], visual sensor networks [5], visual power patrol [6], and optical microscopy [7].

Multi-focus image fusion methods are mainly divided into traditional methods and deep learning methods. Traditional methods usually utilize image processing and computational geometry techniques, among others, while deep learning methods employ deep neural networks, such as convolutional neural networks, for implementation. Since 2017, image fusion technology has experienced explosive development thanks to deep learning-related algorithms, and the research into deep fusion methods has entered a new phase. Researchers from around the world have proposed a large number of multi-focus image fusion methods based on deep learning, which has led to new trends in this field.

Currently, cutting-edge advances in the field of multi-focus image fusion that utilize deep learning are mainly focused on optimizing model structures, applying data enhancement techniques, and designing loss functions. In recent years, researchers have proposed many CNN-based models to improve the efficacy of multi-focus image fusion. To reduce the number of parameters and computational complexity, deep separable convolutional networks have been widely adopted in this area. In the realm of multi-focus image fusion, data enhancement techniques can increase the diversity of training datasets and improve the generalization ability of models. Various new data enhancement techniques have been proposed by researchers, including using generative adversarial networks and methods based on transfer learning, to further enhance the results of multi-focus image fusion. Moreover, several new loss functions have emerged, such as hybrid loss functions that integrate perceptual loss with gradient loss, to further refine the outcomes of multi-focus image fusion.

Image fusion algorithms had been intensively studied before the popularization of deep learning. Early methods for achieving multi-focus image fusion can generally be divided into two categories: spatial domain-based methods and transform domain-based methods. These rely on relevant mathematical transformations to manually analyze the activity levels and to design fusion rules in the spatial or transform domains, and are known as traditional image fusion methods. Specifically, spatial domain-based methods operate directly in the spatial domain and can be further divided into pixel-based methods [8–11], region-based methods [12–15], and block-based methods [16–20]. On the other hand, transform domain-based methods first convert the image to another domain, then utilize the transformed coefficients for fusion, and ultimately obtain the fused image by applying the corresponding inverse transformation. To date, many transform domain-based methods have been proposed, which include sparse representation methods [21, 22], multiscale methods [23, 24], gradient domain-based methods [25], and hybrid methods [26, 27].

However, traditional image fusion methods have numerous limitations. On one hand, to ensure the feasibility of subsequent feature fusion, traditional methods must apply the same transformation to different source images to extract features. Unfortunately, this approach does not account for the various feature differences among the source images, which may result in a poor representation of the extracted features. On the other hand, traditional feature fusion strategies are rudimentary, and as a result, the fusion performance is significantly constrained.

In recent years, researchers and scholars have started introducing deep learning into image fusion to overcome the limitations of traditional image fusion methods. The first study in the

field of multi-focus image fusion using a convolutional neural network model was conducted by Liu et al. [28]. A large-scale dataset, consisting of clear-to-blurry patch pairs generated using high-quality image blocks and their blurred versions, was created. They then used a deep convolutional neural network to encode the mapping in which the focused image integrates clarity information from all source images. In another study, Amin-Naji et al. [29] proposed a novel multi-focus image fusion algorithm based on integrated learning. This algorithm overlaps and integrates the feature information extracted from the input multi-focus images to construct a decision map, enhancing the quality of the output image. Next, Yang et al. [30] presented a multi-level features convolutional neural network (MLFCNN) structure for multi-focus image fusion. In this structure, all the features learned by the network from the previous layer are passed on to the next layer. A $1 \times 1$ convolutional module is added to each pathway to reduce redundancy, and the final multi-focus fusion image is obtained using a pixel-by-pixel weighted averaging strategy. Continue the evolution in this area, Tang et al. [2] proposed a pixel-wise convolutional neural network (P-CNN) model. The P-CNN can accurately measure the focus level of each pixel in the source image. This measurement can effectively avoid artifacts in the fused image, making the model suitable for the problem of multi-focus image fusion. Follow this, Tong et al. [31] suggested an improved dual-channel pulse coupled neural network (IDC-PCNN) model. this IDC-PCNN model can overcome some defects of the standard P-CNN model. Then, Guo et al. [32] introduced a fully convolutional network (FCN) that learns from synthesized multi-focus images., Guo et al. [33] were also inspired by the generative adversarial network (conditional generative adversarial network, GAN) to propose a novel Fuse generative adversarial network (Fuse-GAN) model for image-to-image multi-focus image fusion method.

Due to the limitations of the convolutional operator in the local receptive field, which can't capture enough feature information, the performance of most multi-focus image fusion methods based on convolutional neural networks is limited. To address this problem, a self-attention mechanism was proposed. Specifically, Guo et al. [34] proposed a twin-self-attention network (SSAN) designed to ensure that the foreground region in the generated focused image is marginally larger than the corresponding object. Wang et al. [35] proposed a new Generative Adversarial Network (GAN)-based multi-focus image fusion algorithm called MFIF-GAN (Multi-Focus Image Fusion Generative Adversarial Network). Their model generates a focus probability map by creating focus maps, which helps to attenuate the focus diffusion effect. Similarly, Ma et al. [36] proposed an unsupervised deep learning model, the Squeeze Excitation and Spatial Frequency (SESF-Fuse), to address the multi-focus image fusion problem. Xu et al. [37] presented a gradient and connected regions-based multi-focus image fusion model (GCF). This solution utilizes the gradient relationship of the source image to narrow down the solution domain and accelerate the convergence rate, allowing the constraint on the number of connected regions to help generate a more accurate binary mask. Furthermore, Xu et al. [38] came up with the concept of a many-to-one mapping between the input source image and the output source image. They designed a new deep learning model for multi-focus image fusion by constructing a fully convolutional neural network into a fully convolutional end-to-end two-stream fusion network.

Given that low-level features can only capture low-frequency content, and high-level features are only effective in capturing high-frequency details, Zhao et al. [39] proposed an improved end-to-end multi-focused image fusion model. This model is based on the natural enhancement method of a deep convolutional neural network (MLCNN) that combines multi-level, deeply supervised visual features, thereby improving the fusion performance. Li et al. [40] proposed an end-to-end modeling approach for multi-focused image fusion using a dual U-shaped network. This approach is used to achieve direct mapping from the multi-

focused image to the fused image. In another study, Li et al. [41] proposed a new deep learning network model for multi-focus image fusion called deep Regression Party Learning (DRPL). This model uses a pixel-to-pixel regression strategy, taking a pair of complementary source images as input and converting them into binary masks. This approach enhances the quality of the boundary region, effectively solving the blurring problem around the focus or defocus boundary. Huang et al. [42] proposed an end-to-end generative adversarial network called Auxiliary Classifier Generative Adversarial Network (ACGAN). This network eliminates the need for manually designing complex activity level measurements and fusion rules, and it can output the fused image directly without any post-processing steps.

In current convolutional neural network-based multi-focus image fusion methods, the same convolutional kernel is used to extract features from all regions of the multi-focus image. However, using the same convolutional kernel may not be optimal for all regions in the image, as it can lead to artifacts in the untextured and edge regions of the fused image. To address this issue, Duan et al. [43] proposed a dynamic convolutional kernel network (DCKN) for multi-focus image fusion. In this model, the input image is used as the conditional region to dynamically generate convolution kernels, and the context-aware convolution kernels are utilized to detect focus information. This approach can accurately adapt to the spatial blur caused by depth and texture changes in multi-focus images. Though effective, existing methods based on deep learning generally have an issue of a large number of parameters, which leads to high time complexity and low fusion efficiency of deep learning models. To tackle this problem, Xiao et al. [44] proposed a multi-focus image fusion deep neural network based on the discrete Tchebichef moment function (Discrete Tchebichef moment-based Net, DTMNet). This network has only one convolutional layer with fixed discrete Tchebichef moment function coefficients, enabling efficient extraction of low-frequency or high-frequency information without learning parameters. Additionally, it consists of three fully connected layers with learnable feature classification parameters.

To emphasize better feature exploration, multi-scale features are used. Li et al. [45] presented the first work in the field of multiscale super-resolution. However, the cross-connected architecture leads to complex models and increases the computational cost. Zhen et al. [46] used the concept of feedback mechanism for SISR. Hu et al. [47] proposed a multi-scale information network for high-frequency detail reconstruction. Lu et al. [48] explored multi-scale residual features for better feature extraction. Recently, Wang et al. [49] explored the notion of sparsity in SISR. Despite the considerable improvement brought by pre-upsampling based SR techniques, dedicated methods that consolidate feature representation and edge enhancement capabilities are needed in a single network. Considering the above problems, in our proposed framework, after each multi-scale attention residual block, a novel up-down sampling projection block (UDP) is used to collect high-frequency information to perform edge patching on the extracted multi-scale features.

Attention mechanisms that focus on feature space or channel correlation are heavily used in the field of image processing. The proposed attention-based SR network performs very well, but it comes at the cost of a large number of parameters. Lei et al. [50] follow the ideas proposed for high-level vision problems and exploit channel and spatial attention mechanisms and show better performance. Due to the effectiveness of the attention block, our proposed method further embeds the combined attention module into the proposed residual block. And our proposed lightweight network architecture can promote an increased understanding of image content with a small number of parameters, and utilize both feature and edge information in a single network.

Although many current focused image fusion methods have achieved good fusion results, they tend to produce partial fusion results with artifacts, white noise, and blurred edges when

processing complex images. To obtain more accurate decision maps and better image fusion quality, we propose a multi-focus image fusion model based on a multi-scale hybrid attention residual network. This network directly generates an intermediate decision map to complete the direct mapping from the source image to the fused image, thereby improving the fusion efficiency. Our main contributions are as follows:

1. A multi-scale hybrid attention residual block is designed to adaptively capture the multi-scale correlation between features. We propose a lightweight progressive multi-scale network architecture to explore hierarchical information with fewer parameters, and design a multi-scale hybrid attention residual block, which adaptively captures the multi-scale correlation between features. This lightweight network architecture can efficiently process image features to obtain high-quality fused images.

2. The Up-and-Down Sampling projection block (UDP) is introduced to patch the edge of the extracted multi-scale features. It enables blocks to be able to efficiently exploit multi-scale edge information without increasing the number of parameters.

3. We use a composite loss function to improve the quality of the decision map generated by the network, and the proposed model achieves better fusion performance in terms of subjective evaluation and higher robustness in the quality of the fused images obtained.

## Residual network theory

The complexity and feature extraction ability of a convolutional neural network model mainly depend on the depth of the network. Most neural networks currently seek to increase the network's depth to improve performance. However, increasing the depth of a network comes with costs and challenges. As the depth increases, the neural network becomes more difficult to train. This issue arises because the existing network parameter optimization algorithms are based on backpropagation algorithms. The error values become highly susceptible to issues such as gradient disappearance or gradient explosion after multiple layers of backpropagation, which makes the model difficult to train. The introduction of the Deep Residual Network [51] (DRCN) has solved the aforementioned training issue associated with increasing network depth, attracting significant attention from researchers. The main idea of the residual network is to improve the efficiency of the gradient information propagation by adding shortcut connections to the non-linear convolutional layers. A residual network is a deep network connected by a series of residual modules. As depicted in Fig 1, the residual module consists of two weight layers and a cross-layer short connection, which is then activated by the ReLU activation function and outputted. Compared with general deep networks, the residual network remains easy to train even with deep network depth. This is primarily due to the short connection that allows for the efficient propagation of gradient information between multiple neural network layers, thus overcoming the training difficulties associated with increasing network depth. Consequently, it has become a common model structure for many deep learning tasks.

## Overall network structure

In general, the model of multi-focus image fusion can take two images with different focus angles for fusion. If the number of images to be fused is greater than two, the final fusion results need to be obtained by following a sequential order fusion method. In this paper, we focus on the case of two different images in focus. The overall flow diagram of the proposed algorithm is shown in Fig 2.

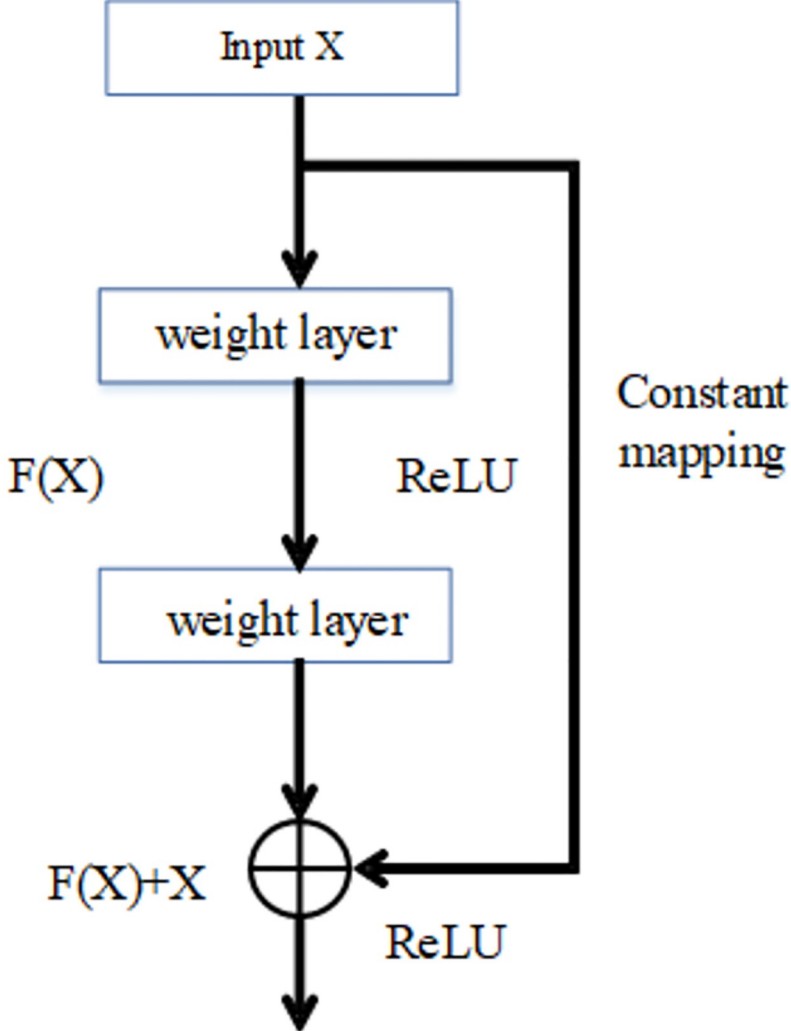

**Fig 1. Individual residual module structure.**

The overall network architecture, as depicted in Fig 2, is primarily divided into three parts. First, the encoder consists of two 3 × 3 convolutional layers. Second, the decoder network comprises four 3 × 3 convolutional layers, serving the primary purpose of reconstructing the input image features. Lastly, the feature fusion module conducts feature fusion on multi-scale

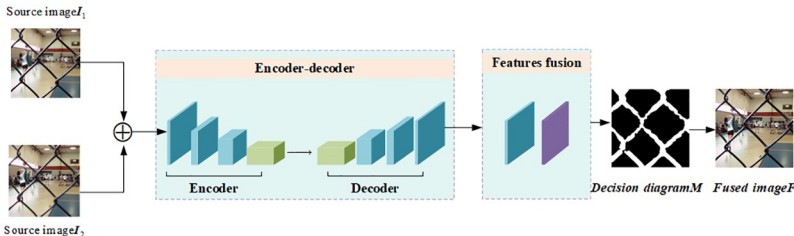

**Fig 2. Overall flow chart.**

feature maps. It merges the recovered feature maps with the source and target images at the pixel level to produce the final fusion results.

First, we use the publicly available MS-COCO 2012 dataset to generate a set of near and far-focused images to train the network model for the algorithm. After the model is trained, the fusion phase is performed by using the network to fuse a set of multi-focus images that are to be processed in order to obtain a clear focus decision map. The generated focus decision map is then optimized using post-processing techniques, such as small region removal and consistency verification methods, to produce the final fusion result map. The principle of the fused image is derived from Eq 1.

$$F = I_1 \times M + (I_2 - M) \tag{1}$$

where $F$ is the fusion result image, $M$ is the decision map, $I_1$, and $I_2$ are the source images.

## Encoder-decoder network structure

In this paper, we mainly use the Encoder-Decoder network to reconstruct the input images throughout the training phase. During the training phase, the Encoder extracts multi-scale features from two source images with different focuses through a Multi-Scale Residual Network (MSRB). It then updates the edge information of the multi-scale features through an Up-and-Down Sampling Projection Block (UDP) module. Subsequently, the extracted features are concatenated and computed to obtain the deep features of the image. The Decoder network consists of four $3 \times 3$ convolutional layers; its primary function is to reconstruct the input image features. In the testing phase, the activity level is calculated by extracting the image features, inputting them into the spatial frequency domain, and obtaining the initial decision map. The decision map is then refined and optimized. The final fused image is obtained by combining the optimized decision map with the source images. The structure of the Encoder-Decoder network is shown in Fig 3.

Our multi-scale residual attention network improves the performance of the model by introducing a multi-scale hybrid attention residual module and a UDP module. The multi-

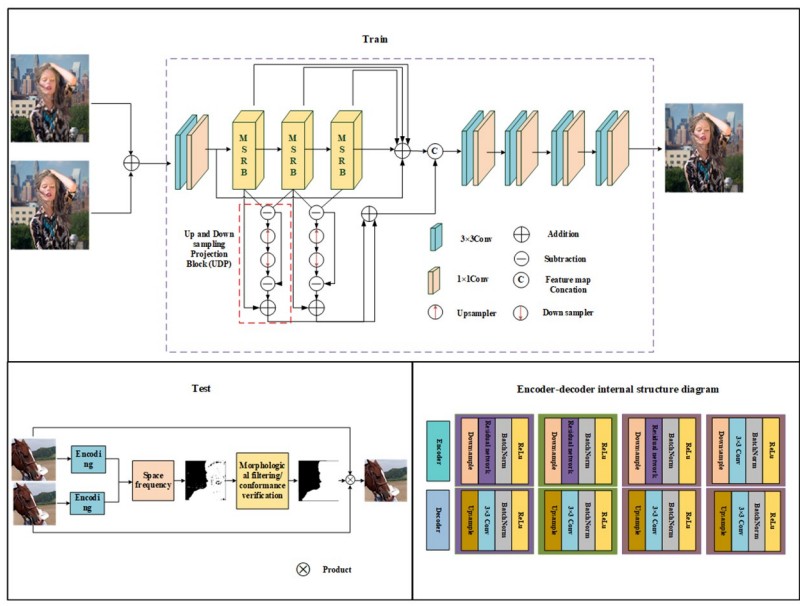

**Fig 3. Overall flow chart.**

scale hybrid attention module includes a hybrid attention mechanism CBAM, which integrates the advantages of channel attention mechanism and spatial attention mechanism, and pays attention to channel features and spatial features at the same time. The attention to the important channels and focus areas of the image was strengthened to improve the feature expression ability of the network. And CBAM has the advantages of lightweight and strong versatility. CAM and SAM in CBAM are lightweight modules, and their internal convolution operations are less, which reduces the amount of calculation and improves the performance of the network under the condition of increasing a small amount of network parameters. In image processing, high-frequency information usually refers to the details or subtle changes in the image that change more frequently. It reflects the edge, texture, detail and high-frequency oscillation information in the image, and the high-frequency information of the image plays an important role in image processing. In the image fusion task, when we perform multi-scale extraction of the image, the resolution of the image is reduced, so some edge information of the image is inevitably lost. The UDP block is used to update the edge of the extracted multi-scale feature to supplement the edge information without increasing the number of parameters.

In our network, MSRB and UDP cooperate with each other to improve the overall performance. We place the MSRB module in the backbone of the network for extracting multi-scale feature representations. However, the UDP module can be used as an auxiliary module to handle the up-and-down sampling operation of the feature map to adapt to the feature requirements of different scales. Through this collaboration, MSRB can make full use of multi-scale feature information for image fusion, while UDP can maintain the consistency and detail expression ability of feature maps. Specifically, MSRB can obtain a more comprehensive and accurate feature representation through multi-scale feature extraction. On the other hand, UDP can maintain the spatial consistency and detail expression ability of the feature map through the up-and-down sampling operation, while reducing the computational complexity. Through the cooperation of these two modules, the performance of the image fusion task can be improved and better fusion results can be obtained.

## Multi-scale mixed attention residuals module

The overall progressive multi-scale model is able to obtain better feature correlation while moving deeper into the network. Unlike the residual blocks proposed in the existing literature [45, 52], our model performs feature extraction more effectively by increasing the perceptual field. To allocate resources more efficiently to the most informative features in the image, we draw inspiration from the attention mechanism described in [53]. To further enhance the network's ability to learn more important features, we designed a hybrid attention unit (CBAM) that combines a spatial attention unit (SAM) with a channel attention unit (CAM). Multi-scale features obtained by aggregating information from parallel convolutional layers of sizes 1, 3, and 5 are connected as follows:

$$\varphi_{msf} = [f_{1\times1}, f_{3\times3}, f_{5\times5}] \tag{2}$$

where $[\cdot]$ and $\varphi$ denote cascaded features and multi-scale features, respectively. Each convolutional layer of the proposed MSRB block consists of 32 filters.

$$X_c = G_{AP}(\varphi_{cfm}) = \frac{1}{h \times w} \sum_{i=1}^{h} \sum_{j=1}^{w} \varphi_{cfm}(i,j) \tag{3}$$

$$\varphi_{CAM} = [\delta(f_{1\times1}(\varepsilon(f_{1\times1}(X_C))))] \times \varphi_{cfm} \tag{4}$$

where $G_{AP}(\cdot)$ represents the global averaging pooling operation that takes into account

channel-level spatial information. $\delta(\cdot)$ represents the sigmoid function, and $\varepsilon(\cdot)$ represents the LeakyReLU activation function. $\phi_{CAM}$ represents the output of the channel attention module. $\phi_{cfm}$ represents the channel feature map, and $X_c$ represents the statistics obtained from the spatial shrinkage $\phi_{msf}$.

To locally regulate the features, a spatial attention unit is used, which is defined as

$$\varphi_{SAM} = (f_{1\times1}(\phi(\varphi_{msf}))) \qquad (5)$$

$$\varphi_{cat} = [\varphi_{SAM} \times \varphi_{msf}, \varphi_{CAM} \times \varphi_{msf}] \qquad (6)$$

$$\varphi_{MSRB} = \varphi_{CABM} = \varphi_{cat} + \varphi_{msf} \qquad (7)$$

where $\phi_{SAM}$ denotes the output of the spatial attention block. $\phi_{MSRB}$ shows the final output of the MSRB block. $\phi(\cdot)$ means that the size of the filter is $3 \times 3$, the number of channels is 3, and the depth convolution operation is performed 3 times. $\phi_{cat}$ represents the connected attentional features. $\phi_{CABM}$ indicates the mixed attention unit. The proposed attention block is able to exploit inter and intraframe channel information, where the use of deep convolution further helps to generate a different 2D spatial attention map for each channel, and then the obtained attention map is better updated by passing it through a convolutional layer with 64 filters. In order to take advantage of both blocks simultaneously, we combine them through channel concatenation, resulting in a hybrid attention unit.

The CBAM integrates the advantages of the channel attention mechanism and the spatial attention mechanism, focusing on both channel features and spatial features. It enhances the attention to important channels and focal regions of images and improves the feature representation capability of the network. Both the CAM and SAM in CBAM are lightweight modules with a reduced number of internal convolution operations, which decrease the computational burden. These modules improve the performance of the network with only a small increase in the number of network parameters.

CBAM integrates the advantages of the channel and spatial attention mechanisms, focusing on both channel features and spatial features simultaneously. This focus enhances attention to the important channels and the focal regions of an image, and improves the network's feature representation ability. CBAM is both lightweight and versatile. The CAM and SAM within CBAM are lightweight modules with fewer internal convolution operations, reducing the computational load. By utilizing two $3 \times 3$ convolution kernels instead of a single $5 \times 5$ kernel, the network's depth is increased while maintaining the same receptive field, thereby improving the performance of the network with only a modest increase in the number of parameters. Multi-scale hybrid 321 residual network structure, as shown in Fig 4.

## Upper and lower projection blocks

After obtaining fine features from the MSRB block, we increase the high-frequency information content in the image using the UDP block, as shown in Fig 3. The overall operation of the UDP block is demonstrated in Eqs 8 and 9. First, the UDP module evaluates the discrepancy of the continuous multi-scale feature map output by the MSRB, focusing on high-frequency information. The features with reduced information are then passed through an upsampling layer, Conv2DTranspose, with a stride of 2. The upsampled features are transformed back into the LR (low-resolution) input space by using a convolution layer with a stride of 2. The final subtraction operation, resulting in the output shown in Eq 8, helps to remove redundant information. The addition operation in Eq 9 extracts the relevant features needed to reconstruct a clear image, thus enhancing the information content of the multi-scale features. Additionally,

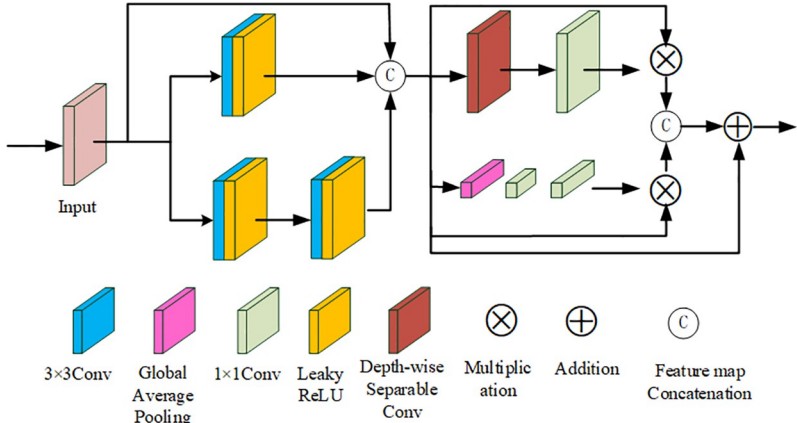

**Fig 4. Multi-scale hybrid attention residual module.**

the features from all UDP blocks are aggregated to promote better gradient propagation. Finally, all residual and edge features are combined to fully utilize the multi-scale edge features.

Here, $\uparrow 2$ and $\downarrow 2$ represent the operations of upsampling and downsampling of $\times 2$, respectively. $\vartheta_n$ represents the de-output features, $i_n$ represents the redundant information, $\delta_n$ shows the features after removing the redundant information, $\varphi_{msf_n}$ and $\varphi_{msf_{n-1}}$ represents the multi-scale information of the $N_{TH}$ and $n-1$ layers.

$$
\begin{aligned}
\delta_n &= \vartheta_n - i_n \\
\vartheta_n &= \varphi_{msf_n} - \varphi_{msf_{n-1}} \\
i_n &= \downarrow 2(\uparrow 2(\vartheta_n))
\end{aligned} \tag{8}
$$

$$
\phi_{UDP} = \delta_n + \varphi_{msf_n} \tag{9}
$$

## Multi-scale feature fusion

In this paper, we develop a dual-bypass fusion system that uses multiple convolutional kernels for various bypasses. This configuration allows the signals between the bypasses to be shared with each other, enabling the fusion of image features of various sizes. The formula is defined as follows:

$$
\begin{aligned}
S_1 &= \sigma(W_{3\times3}^1 * M_{n-1} + b^1) \\
P_1 &= \sigma(W_{5\times5}^1 * M_{n-1} + b^1) \\
S_2 &= \sigma(W_{3\times3}^2 * [S_1, P_1] + b^2) \\
P_2 &= \sigma(W_{5\times5}^2 * [P_1, S_1] + b^2) \\
S' &= (W_{1\times1}^3 * [S_2, P_2] + b^3)
\end{aligned} \tag{10}
$$

where $W$ and $b$ denote the weight and bias values, the superscript denotes the number of layers they are in, and the subscript denotes the size of the convolution kernel used in that layer. $\sigma(x) = \max(0, x)$ represents the ReLU function and $[S_1, P_1]$, $[P_1, S_1]$, $[S_2, P_2]$ represents the join operation. $S'$ represents the fused image features.

The specific operation is that the multi-scale feature map is input into the double bypass system, and the double bypass system, for each scale feature map, inputs it into two independent bypasses. A variety of convolution calculation kernels are used for each bypass, so that the information between their side channels can be shared with each other. In each bypass, feature fusion is performed on the processed feature map. We assign a weight to the feature maps at each scale and add them by weight. The feature map is then resized to the same size as the original image using operations such as upsampling. The recovered feature maps can then be pixel-wise fused with the source and target images to generate the final fusion result.

## Loss function design

Structural similarity function loss $L_{ssim}$ and pixel function loss $L_p$ are used as loss functions in many computer vision tasks to optimize method parameters. However, using them alone to constrain the fusion result is not enough to further improve the quality of the fused image. Therefore, our researchers try to use a composite function combining multiple loss functions to better optimize the network. Because multi-scale features need to be extracted and fused in our proposed method, we add the feature loss loss function to evaluate the quality of the fused image from the feature dimension. We use a novel composite loss function to complete end-to-end training without the help of additional trained models in the implementation of feature loss terms. I adopt a joint loss function $L_{all}$ consisting of three parts to optimize the proposed structure, which can be expressed in Eq 11:

$$L_{all} = \lambda L_{ssim} + \mathrm{L}_p + \beta L_{per} \tag{11}$$

Where $\lambda$ and $\beta$ denote the weighting factor of the $L_{ssim}$ loss term and the $L_p$ loss term, respectively. According to the experimental experience, we set $\beta = 0.1$ and $\lambda = 3$. $L_{ssim}$ represents the multiscale structural similarity loss term. The pixel loss $L_p$ is the Euclidean distance between the output $O$ and the input $I$ of the network:

$$L_p = \|O - I\|_2 \tag{12}$$

The structural similarity function loss $L_{ssim}$ represents the structural difference between the input and output.

$$L_{ssim} = 1 - \mathrm{SSIM}(O, I) \tag{13}$$

where SSIM represents the structural similarity operation. The expressions for luminance contrast and structural similarity are shown in Eq 14, where $\mu$ is the mean, $\sigma$ is the standard deviation, and $C_1$, $C_2$, and $C_3$ are constants.

$$\mathrm{SSIM}(O, I) = \frac{2\mu_O\mu_I + C_1}{\mu_O^2 + \mu_I^2 + C_1} \times \frac{2\sigma_O\sigma_I + C_2}{\sigma_O^2 + \sigma_I^2 + C_2} \times \frac{\sigma_{O,I} + C_3}{\sigma_O^2\sigma_I^2 + C_3} \tag{14}$$

## Fusion results and analysis

### Experimental data set and preprocessing

The MS-COCO2012 dataset, consisting of over 120,000 images, is used for this experiment. In this study, the dataset is divided into a training set with 104527 images, a test set with 10286 images, and a validation set with 10725 images. The validation set primarily serves to verify network convergence during the training process. The images are resized to $256 \times 256$ and converted to grayscale. We set the learning rate at $1 \times 10^{-4}$, with a batch size of 16 and 300

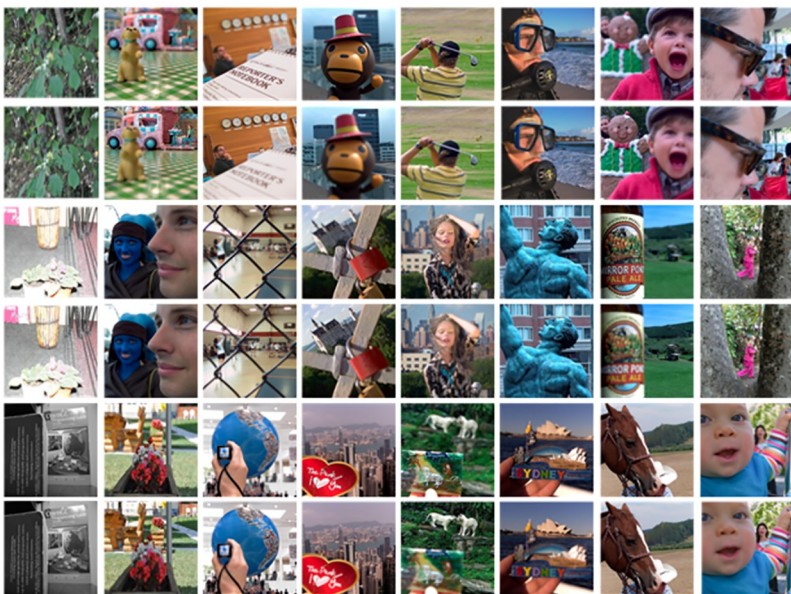

**Fig 5. The multifocal image test dataset.**

training rounds. Since the training parameters are fixed, we test the fused images on the test set, part of which is shown in Fig 5. For the experiment, we select 100 sets of image data as test images. The "Lytro" multi-focus image dataset, the most commonly used public dataset, contains images of size 520 × 520 and includes both color and grayscale images.

## Experimental environment and parameter settings

The proposed deep learning-based multi-focus image fusion algorithm is implemented using the Python programming language and the PyTorch deep learning framework. The hardware environment consists of a GTX 1080Ti GPU with 16GB of video memory.

## Quality evaluation index

It is difficult for us to make an accurate evaluation of the fusion performance only by subjective evaluation, so fusion indicators are also needed for objective evaluation. Various fusion metrics have been proposed in many studies, but none of them seems to be absolutely superior to the others. Therefore, it is necessary to select multiple metrics to evaluate different fusion methods.

This paper uses a variety of evaluation metrics, such as average gradient (*AVG*), which mainly reflects the change of image resolution and texture. With the increase of the gradient, the higher the resolution of the image, the clearer the image. Standard deviation (*STD*), the fused image with high contrast tends to have a larger standard deviation measure, which means that the fused image will obtain better visual effects. Structural Similarity index (*SSIM*), which tests the distortion contrast and structural similarity loss of the source image and the fused image, and a higher value indicates better fusion performance. Peak Signal-to-Noise Ratio (*PSNR*), which is a metric that reflects the distortion by the ratio of peak power to noise power, the higher the value, the better the fusion performance. Visual information fidelity (*VIF*), which detects the fused image and evaluates the quality and performance of the

evaluated image through the correlation between the evaluated image and the reference image. Mutual information ($Q_{NMI}$), which indicates how much edge information the fused image obtains from the source image, the larger its value is, the more edge information exists in the fused image. Based on human visual perception index ($Q_{CB}$), in information theory, it is a measure of correlation between two random numbers of an important indicator, measure of transfer of information from the source image and the fusion image. Edge information retention index ($Q_{AB/F}$), which is a quantitative evaluation index of fused image quality based on human visual perception. From the perspective of human visual system, it evaluates the fusion by comparing the contrast feature information between the image to be fused and the fused image. Based on structural similarity index $Q_Y$, it is a kind of based on image structure similarity metric, you can evaluate how much $F$ retained the fused image from the source image $A$ and $B$ structure information.

## Comparative analysis

On the "Lytro" dataset, we compare our method with those of other representative algorithms. These include the curvilinear wavelet transform (CWT) [54], the dual-channel pulse-coupled neural network (IDCP-CNN) [55], dense SIFT [56], multi-scale weighted gradient (MWG) [57], multi-focus image fusion generative adversarial network (MFIF-GAN) [58], and Squeeze-and-Excitation and Spatial Frequency fusion (SESF-Fuse) [59], as well as Fine-grained Multi-focus Image Fusion (FGMF) [60]. Multi-focus image fusion using structure-guided flow (MFST) [61]. The quantitative evaluation results of the different algorithms, analyzed through experimental comparison, are shown in Table 1.

Through experimental comparison and analysis, our algorithm demonstrates the best performance on six different quantitative metrics. The results indicate that the proposed algorithm can more effectively preserve the pixel information of the source images. Moreover, all the information related to pixel structure and edge contours from the source images is well transferred to the fused image. Additionally, the network enhances the visual effect in a manner consistent with human visual perception.

The results of fusing this algorithm with the Lytro-01 dataset are shown in Fig 6. It can be seen that CWT, IDCP-CNN, DSIFT, MWG, and SESF-Fuse had many unclear areas after the pseudo-color technique was applied, mainly in the far-focused images. The pseudo-color image of SESF-Fuse in the far-focused region exhibited significant white noise and a blurred background. Similarly, the pseudo-color images of FGMF and MFIF-GAN in the far-focused regions contained considerable white noise around the focused and unfocused edges, and the background image was blurred, indicating that the entire area could not be detected as a

**Table 1. Quantitative experimental results of the present algorithm and other methods.**

| Approachs | AVG | STD | PSNR | SSIM | VIF | QAB/F |
|---|---|---|---|---|---|---|
| CWT | 8.2375 | 61.6748 | 26.8376 | 0.8342 | 0.5760 | 0.6856 |
| MWG | 8.2661 | 62.2006 | 27.3152 | 0.8248 | 0.6671 | 0.7109 |
| DSIFT | 8.31235 | 62.3732 | 27.1251 | 0.8185 | 0.7106 | 0.7352 |
| IDCP-CNN | 8.1590 | 62.2523 | 27.1488 | 0.8315 | 0.7142 | 0.7326 |
| MFIF-GAN | 8.2453 | 62.2615 | 27.2337 | 0.8315 | 0.7146 | 0.7315 |
| SESF-Fuse | 8.3010 | 62.3140 | 27.2590 | 0.8316 | 0.7152 | 0.7321 |
| FGMF | 8.3114 | 62.4008 | 27.2775 | 0.8384 | 0.7318 | 0.7561 |
| MFST | 8.3189 | 62.3982 | 27.2854 | 0.8395 | 0.7359 | 0.7556 |
| **Ours** | **8.3475** | **62.4354** | **27.3391** | **0.8415** | **0.7490** | **0.7576** |

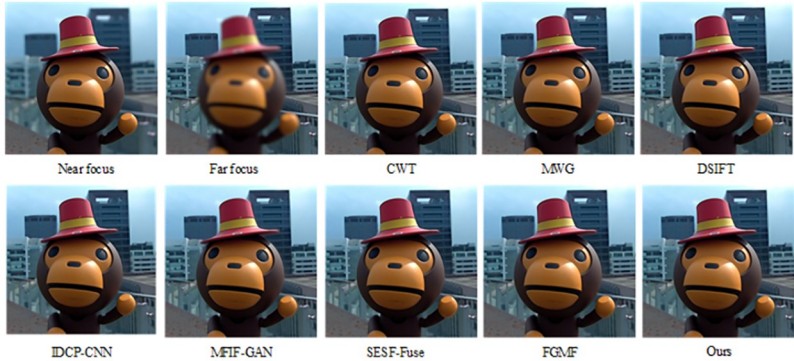

**Fig 6. Fusion results of the proposed algorithm and other methods on Lytro-05.**

focused region. From Fig 7, it is apparent that the algorithm discussed in this paper can completely segment and clearly detect focused and unfocused edge information at the far focus.

The fusion results of our algorithm and those of other representative algorithms on the Lytro-05 dataset are shown in Fig 8. From the pseudo-color image in Fig 9, it is apparent that CWT, MWG, DSIFT, and IDCP-CNN produce a large number of artifacts on the right side of the baby's cheek in the focused image, indicating incomplete image segmentation. It can also be seen that MFIF-GAN presents obvious artifacts in focus, while SESF-Fuse exhibits both artifacts and fuzzy contours at focus, as well as residual pixels. In the case of FGMF, there is some white noise in the periocular area of the infant within the red box, but the overall pseudo-color image is better. As demonstrated by the pseudo-color image in Fig 9, our algorithm effectively addresses these issues, achieving a good fusion map and pseudo-color image.

The results of fusing our algorithm with the Lytro-05 dataset are shown in Fig 10. As can be seen, CWT, MWG, DSIFT, IDCP-CNN, and MFIF-GAN leave many areas unclear after pseudo-color processing, indicating incomplete image segmentation. The SUSE-Fuse algorithm presents jagged residuals in the red box and also shows unclear segmentation at the boundary. The pseudo-color image of FGMF in the far-focus region has a small amount of

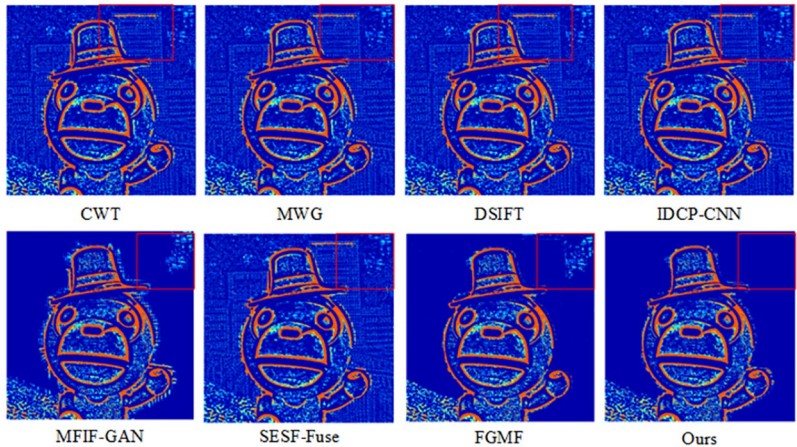

**Fig 7. Pseudo-color results of our algorithm versus other methods on Lytro-05.**

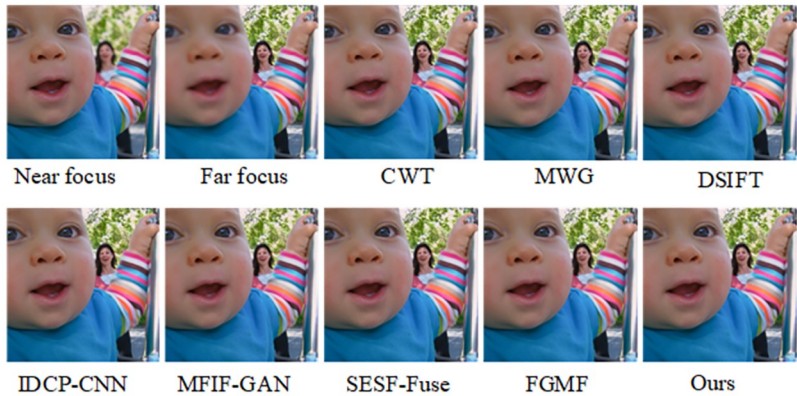

**Fig 8. Fusion results of the proposed algorithm and other methods on Lytro-01.**

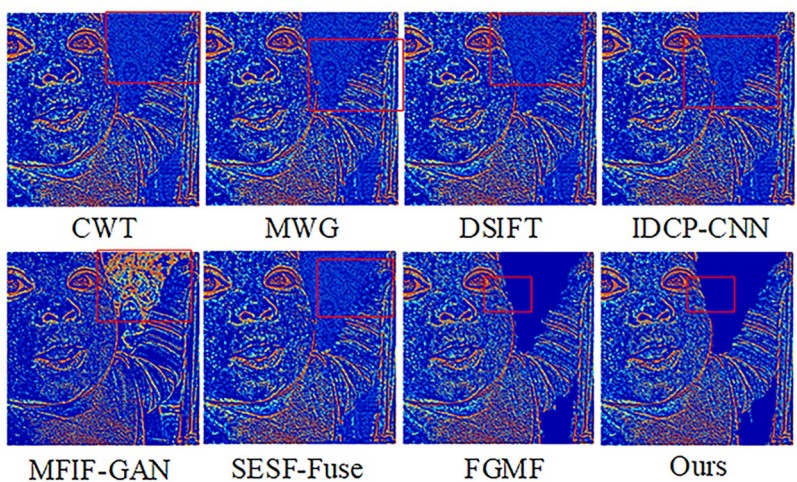

**Fig 9. Pseudo-color results of our algorithm versus other methods on Lytro-01.**

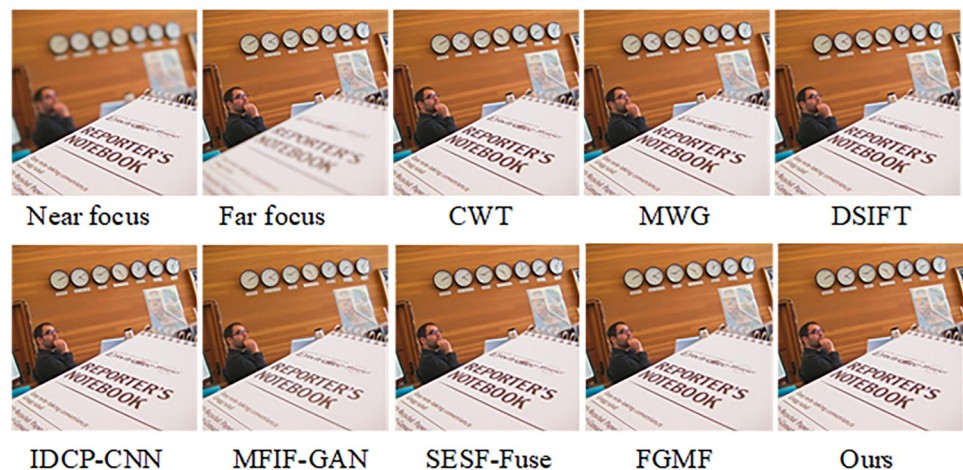

**Fig 10. Fusion results of the proposed algorithm and other methods on Lytro-05.**

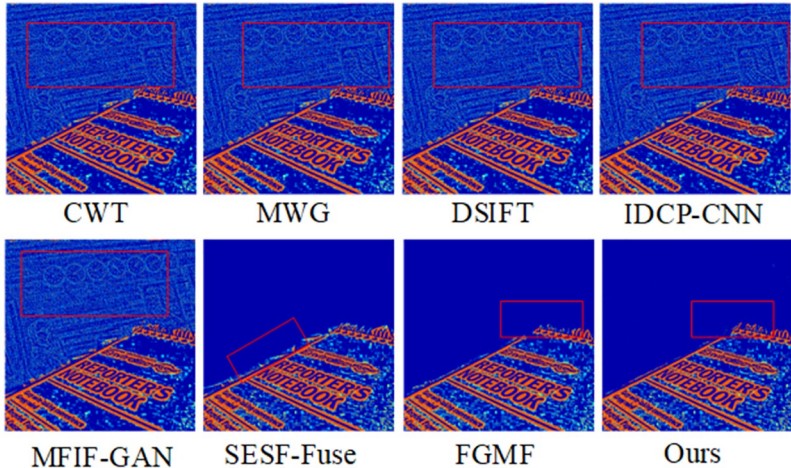

**Fig 11. Pseudo-color results of our algorithm versus other methods on Lytro-05.**

white noise and image blur in the focus/non-focus boundary area, suggesting that this area cannot be entirely detected as focused. From the pseudo-color images corresponding to each algorithm in Fig 11, it is evident that our algorithm can fully segment the far-focus region and provides clear detection of the focused and unfocused edge information.

The results of the fusion of our algorithm with the Lytro-05 dataset are presented in Fig 12. This figure reveals that the pseudo-color processing by CWT, MWG, DSIFT, IDCP-CNN, and MFIF-GAN leads to numerous unclear areas, indicating that image segmentation is not thorough. The SUSE-Fuse algorithm produces a jagged residual effect on the man's glasses, as well as unclear segmentation at the boundaries. In the far-focus image, the FGMF pseudo-color image exhibits a small amount of white noise and image blur in the region between the focused and unfocused edges, suggesting that this region cannot be fully detected as a focused area. As shown by the pseudo-color images corresponding to each algorithm in Fig 13, the proposed algorithm is capable of completely segmenting the far-focus area, and it clearly detects the information at the edges of the focused and unfocused areas.

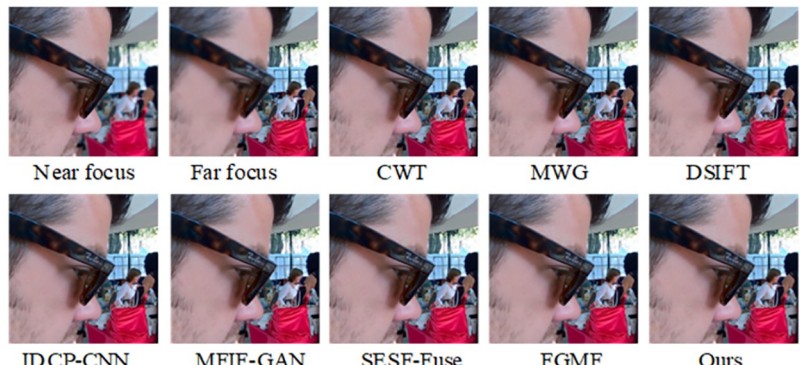

**Fig 12. Fusion results of the proposed algorithm and other methods on Lytro-05.**

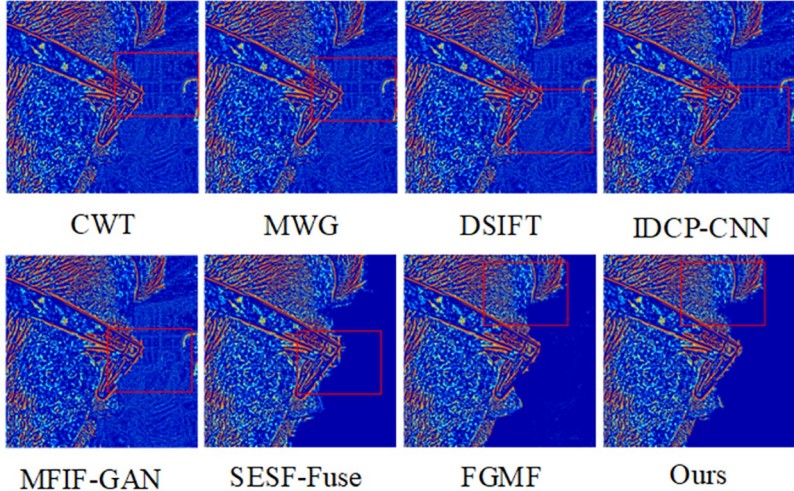

**Fig 13. Pseudo-color results of our algorithm versus other methods on Lytro-05.**

## Visualization of fusion results

As shown in Fig 14, the first row is the foreground image, the second row is the far-field image, the third row is the gradient image, and the last row is the result image. The experimental results show that the fused image was clearer and better.

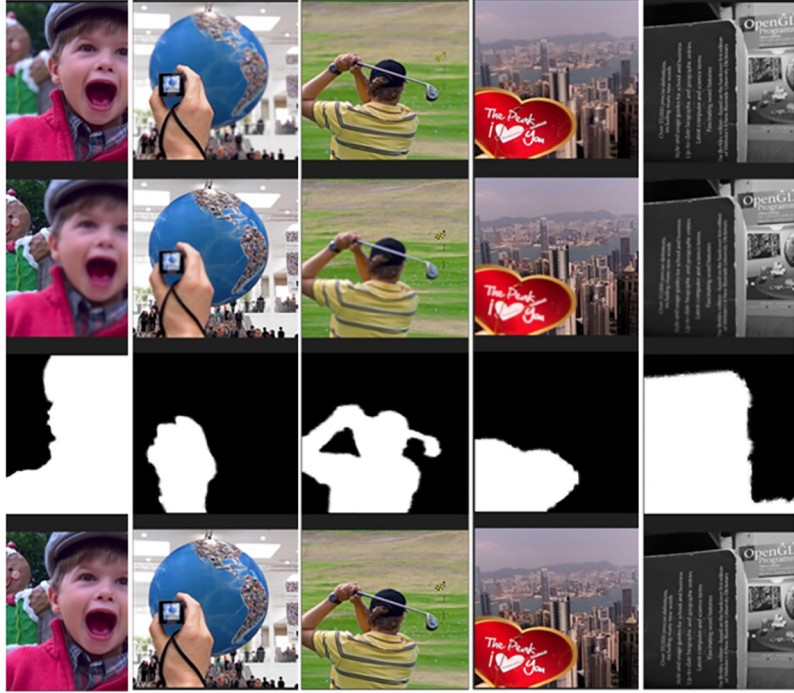

**Fig 14. Visualization of the fusion results.**

**Table 2. Experimental results of quantitative ablation of network structure.**

| Algorithm | $Q_{NMI}$ | $Q_{AB/F}$ | $Q_Y$ | $Q_{CB}$ |
|---|---|---|---|---|
| No/MSRN | 1.02366 | 0.65380 | 0.952341 | 0.752342 |
| MSNR | 1.05536 | 0.704531 | 0.958641 | 0.767840 |
| MSNR+UDP | 1.07512 | 0.710124 | 0.961512 | 0.778106 |
| 2MSNR+UDP | 1.09542 | 0.714579 | 0.968551 | 0.798552 |
| **Ours** | **1.12458** | **0.734595** | **0.975871** | **0.814675** |

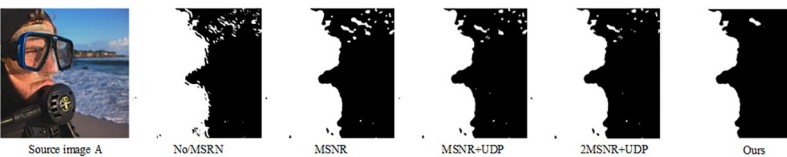

**Fig 15. Focusing decision diagrams generated by different network structure.**

## Module ablation experiment

In order to verify the effectiveness of the above two important components of the network structure, experiments were conducted on the "Lytro" dataset by comparing the two new modules obtained by adding and removing the modules of the algorithm in this paper. In particular, we perform ablation experiments on a single multi-scale residual block and a multi-scale residual block plus a UDP module, along with a multi-scale residual block and an attention mechanism. The algorithm was divided into five different modules, and the analysis of the experimental results shows that the algorithm in this paper achieved the highest score with stronger fusion effect and also improved the quality of the fused images. The quantitative evaluation results and the focus decision diagram were shown in Table 2 and Fig 15.

Where $Q_{NMI}$ represents normalized mutual information, $Q_{AB/F}$ represents edge information retention, $Q_Y$ represents structural similarity, and $Q_{CB}$ represents human visual perception.

## Loss function ablation experiment

To verify the effectiveness of the above two important components of the loss function, the loss function of this algorithm was compared with the control variables method and the experimental analysis was conducted on the "Lytro" dataset. The experimental results shows that our algorithm achieved the highest score, which indicates that our algorithm had stronger fusion effect and the quality of fuse images was improved. The quantitative evaluation results and focus decision diagrams were shown in Table 3 and Fig 16.

**Table 3. Experimental results of quantitative loss function ablation.**

| Algorithm | $Q_{NMI}$ | $Q_{AB/F}$ | $Q_Y$ | $Q_{CB}$ |
|---|---|---|---|---|
| Only Lp | 0.97567 | 0.69880 | 0.94357 | 0.75861 |
| Only Lssim | 1.00579 | 0.66423 | 0.97552 | 0.74358 |
| Only Lper | 1.01024 | 0.64231 | 0.95276 | 0.75018 |
| **Ours** | **1.35312** | **0.77473** | **0.98625** | **0.77881** |

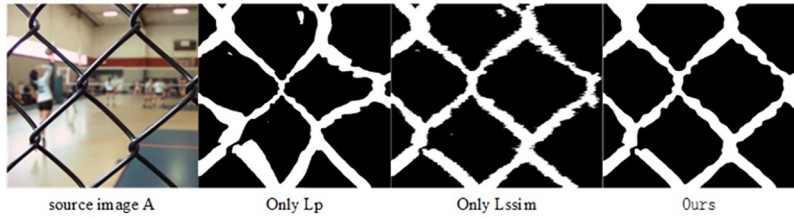

**Fig 16. Focusing decision diagrams generated by different loss functions.**

Where $Q_{NMI}$ represents normalized mutual information, $Q_{AB/F}$ represents edge information retention, $Q_Y$ represents structural similarity, and $Q_{CB}$ represents human visual perception.

### The optimal choice of MSRB and UDP

We use different numbers of MSRB modules to further study the performance of the proposed network, as shown in Table 4, with the increase of the number of multi-scale MSRB modules, each performance index is increasing, indicating that the performance of the network becomes better, but when the number of MSRB modules is more than 3, the improvement of network performance can be ignored. So we used 3 MSRB modules to ensure better feature learning. To verify the importance of the UDP module, we analyzed the performance of the network with and without the UDP module, and Table 4 shows the improvement of the network performance when using the UDP module.

### Model feasibility analysis

We analyze and compare the time efficiency, computational complexity and floating-point operation of the model. Model feasibility analysis is shown in Table 5. It can be seen that the computational complexity of our proposed model is the lowest, the floating point number is relatively small, and the average fusion time is also the shortest, which verifies the feasibility of our proposed model.

### Multi-aggregate image sequence fusion verification

The actual scenario of multi-focus image fusion studied in this paper is the fusion of two original images. In order to reflect the applicability and feasibility of the algorithm in this paper, the scattered focus images of three positions in the same scene are taken for their fusion. The first step was to fuse the first two different location images, and then fused the result image with the last image of different angles. In this paper, the scatter images of three different locations in the same scene from the "Lytro" dataset were used for image fusion. The fusion results were

**Table 4. Optimal selection of the number of MSRB and UDP.**

| Number of MSRB | UDP | $Q_{NMI}$ | $Q_{AB/F}$ | $Q_Y$ | $Q_{CB}$ |
|---|---|---|---|---|---|
| 1 | - | 1.05456 | 0.704501 | 0.958443 | 0.767245 |
| 2 | - | 1.07563 | 0.712213 | 0.961321 | 0.775430 |
| 3 | without | 1.07831 | 0.720458 | 0.967249 | 0.779534 |
| 3 | with | 1.12436 | 0.734684 | 0.975564 | 0.814475 |
| 4 | without | 1.07871 | 0.720658 | 0.967549 | 0.779834 |

**Table 5. Model feasibility analysis.**

| Approach | Parameters | Flops | Times |
|---|---|---|---|
| CWT | 19.7M | 182.5G | 13s |
| MWG | 22.5M | 108.2G | 18s |
| DSIFT | 38.5M | 118.3G | 29s |
| IDCP-CNN | 40.1M | 205.1G | 35s |
| MFIF-GAN | 15.4M | 92.5G | 10s |
| SESF-Fuse | 10.1M | 42.1G | 7s |
| FGMF | 8.3M | 50.6G | 20s |
| MFST | 12.6M | 60.2G | 4.1s |
| Our | 10.5M | 55.6G | 1.5s |

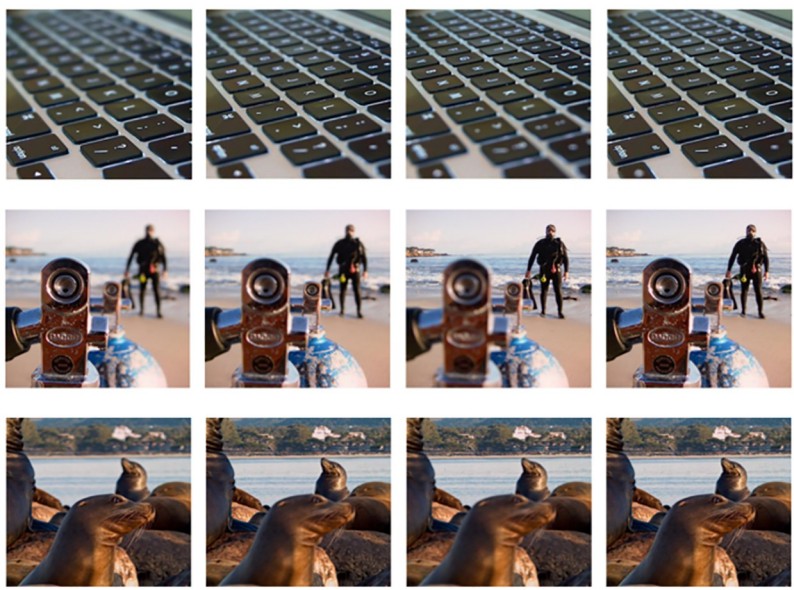

1) Source image   2) Source image   3) Source image   4) Fusion image

**Fig 17. Multi-angle image fusion results.**

shown in Fig 17. The experimental results showed that a complete fused image can be obtained by experimenting on three different focus angle images.

## Conclusions

In this paper, we propose a multi-focus image fusion algorithm based on a multi-scale hybrid attention residual network. In order to detect the information of the source image completely and accurately and obtain a high-quality focused decision map, this paper innovates from two aspects of network structure and loss function respectively. First, the method uses the Encoder-decoder network, while introducing the multi-scale mixed attention residual block into the encoder to obtain the focusing characteristics of the source image. Then, it extracts edge information of the multi-scale features through UDP. Subsequently, the multi-scale

feature information and multi-scale edge information are input into the decoder to reconstruct the feature image. Finally, the multi-scale feature fusion module is used for feature fusion to obtain the fused image. Second, a hybrid loss function of $L_p$ and $L_{ssim}$ is introduced in the design of the loss function to improve the quality of the decision graph generated by the network. The advantages of this paper's algorithm in multi-aggregate image fusion are verified by objective index evaluation and subjective visual comparison. The ablation experiments on the network structure and loss function prove the feasibility of the algorithm design and innovation of this paper, which has practical application value.

## Author Contributions

**Conceptualization:** Tingting Liu, Mingju Chen.

**Data curation:** Tingting Liu, Anle Cui.

**Formal analysis:** Mingju Chen, Zhengxu Duan.

**Investigation:** Tingting Liu, Anle Cui.

**Project administration:** Tingting Liu, Mingju Chen.

**Resources:** Tingting Liu.

**Software:** Tingting Liu, Zhengxu Duan.

**Supervision:** Tingting Liu.

**Validation:** Tingting Liu, Mingju Chen.

**Visualization:** Tingting Liu.

**Writing – original draft:** Tingting Liu.

**Writing – review & editing:** Zhengxu Duan, Anle Cui.

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
