## [Decision Letter · Decision Letter 0]

30 Jan 2024

PONE-D-23-44021Multi-focused image fusion algorithm based on multi-scale hybrid attention residual networkPLOS ONE

Dear Dr. Liu,

Thank you for submitting your manuscript to PLOS ONE. After careful consideration, we feel that it has merit but does not fully meet PLOS ONE’s publication criteria as it currently stands. Therefore, we invite you to submit a revised version of the manuscript that addresses the points raised during the review process.

We look forward to receiving your revised manuscript.

Kind regards,

Khan Bahadar Khan, Ph.D

Academic Editor

PLOS ONE

Journal Requirements:

"This research was funded by Natural Science Foundation of Sichuan, China(2023NSFSC1987,2022ZHCG0035); The Key Laboratory of Internet Information Retrieval of Hainan Province Research Found(2022KY03); the Opening Project of International Joint Research Center for Robotics and Intelligence System of Sichuan Province(JQZN2022-005); Sichuan University of Science & Engineering Postgraduate Innovation Fund Project, grant number Y2022130."

6. Please upload a copy of Figures 1 to 17, to which you refer in your text on pages 5-7 and 9-12. If the figure is no longer to be included as part of the submission please remove all reference to it within the text.

8. We are unable to open your Supporting Information file [Image of the article.rar]. Please kindly revise as necessary and re-upload.

Reviewers' comments:

Reviewer's Responses to Questions

**Comments to the Author**

1. Is the manuscript technically sound, and do the data support the conclusions?

Reviewer #1: Yes

Reviewer #2: Partly

Reviewer #3: Yes

2. Has the statistical analysis been performed appropriately and rigorously? 

Reviewer #1: Yes

Reviewer #2: Yes

Reviewer #3: N/A

3. Have the authors made all data underlying the findings in their manuscript fully available?

Reviewer #1: No

Reviewer #2: No

Reviewer #3: No

4. Is the manuscript presented in an intelligible fashion and written in standard English?

Reviewer #1: Yes

Reviewer #2: Yes

Reviewer #3: Yes

5. Review Comments to the Author

Reviewer #1: The current work proposed the multi-focus image fusion network based on deep learning for improving the detection. Some comments need to be addressed before its publication:

How to weight the two different components in the loss function needs to be further explained and demonstrated.

The evaluations of Tables 2 and 3 need more explanations especially on these quantitative paramters demonstrated.

Reviewer #2: The paper proposes a multi-focused image fusion algorithm based on multi-scale

hybrid attention residual network. Though I found the contribution very interesting, the following problems need to be addressed.

Questions and observations are summarized (not in order of importance) in the following items:

1. The content of this paper is incomplete, all figures are not provided.

2. Please cite relevant works from the past 3 years in the literature section.

3. in formulas 3 and 4, the symbols used differ from those appearing in the formulas，for example，ϕCAM and φCAM. It is suggested to change for consistency to facilitate readers understanding.

4. Why didn't the symbol ϵ(·) below formula 4 and the symbol ϕ below formula 10 appear in both formulas?

5. What does S' mean in formula 10? Please provide an explanation below formula 10 for better understanding by the readers.

6. Is Lsimm in contribution 3 the same as Ls in formula 13? If not, please provide clarification.

7. Eight methods were proposed in the comparative analysis, but only seven are presented in Table 1. Is the GF-IMF method not included for comparison?

8. The ablation study is insufficient. The authors only provide the ablation experiment of Loss function.

9. All parameters in formulas should be described and explained.

10. Lp and Lsimm are the classical loss function. In view of this point, the contribution 3 can not be regarded as the innovation.

Reviewer #3: Here are my detailed critiques:

1. The paper claims robust performance across various datasets but lacks rigorous testing on diverse, real-world scenarios. The primary reliance on the MS-COCO2012 dataset raises concerns about the model's ability to generalize to different types of multi-focus images, especially in more complex or less controlled environments.

2. While the paper introduces MSHRN, it does not sufficiently differentiate this approach from existing methods. The innovation in combining multi-scale features and attention mechanisms needs to be better highlighted, or the incremental nature of this improvement should be more transparent.

3. The role and effectiveness of the Up-and-Down Sampling Projection (UDP) module are not convincingly established. How does this module specifically contribute to the network's performance, especially in comparison to other edge-enhancement techniques?

4. While the paper includes comparative analysis with several methods, the selection of these comparative algorithms appears limited. Inclusion of more recent and diverse state-of-the-art methods would better position the paper's contributions within the current research landscape.

5. The reliance on subjective evaluation is a significant drawback. More rigorous quantitative metrics are needed to objectively assess the performance of the proposed method. Also, the manuscript does not provide a clear rationale for the choice of the specific metrics used.

6. The paper lacks detailed explanation and justification for the choice of network architecture, including the number of layers, filter sizes, and specific configurations of the MSHRN and UDP modules. Such details are crucial for reproducibility and critical assessment.

7. The combination of structural similarity function loss (Ls) and pixel function loss (Lp) is not novel. A more thorough justification for this specific combination and its parameter tuning is needed, including why this combination is more effective than others.

8. The paper does not provide sufficient theoretical grounding for the proposed method, particularly in how the MSHRN and UDP modules contribute to the overall performance improvement. More theoretical analysis or insights into why this architecture works effectively for multi-focus image fusion is needed.

9. The manuscript does not adequately address the computational efficiency of the proposed method, which is a crucial factor in real-world applications. Information about the model's computational complexity, memory requirements, and execution time would provide a more comprehensive assessment of its practical viability.

6. PLOS authors have the option to publish the peer review history of their article (what does this mean?). If published, this will include your full peer review and any attached files.

Reviewer #1: No

Reviewer #2: No

Reviewer #3: No

---

## [Author Response · Author response to Decision Letter 0]

23 Mar 2024

Dear Editor,

1 Our manuscript was revised according to the style template of PLOS ONE.

2. We have modified and confirmed the funding information.

3.Funders' Role in this study: Funders had no role in study design, data collection and analysis, publication decisions, or manuscript preparation. We included the relevant content in our cover letter

---

## [Decision Letter · Decision Letter 1]

9 Apr 2024

Multi-focused image fusion algorithm based on multi-scale hybrid attention residual network

PONE-D-23-44021R1

Dear Dr. Liu,

We’re pleased to inform you that your manuscript has been judged scientifically suitable for publication and will be formally accepted for publication once it meets all outstanding technical requirements.

Kind regards,

Khan Bahadar Khan, Ph.D

Academic Editor

PLOS ONE

Additional Editor Comments (optional):

Reviewers' comments:

Reviewer's Responses to Questions

**Comments to the Author**

1. If the authors have adequately addressed your comments raised in a previous round of review and you feel that this manuscript is now acceptable for publication, you may indicate that here to bypass the “Comments to the Author” section, enter your conflict of interest statement in the “Confidential to Editor” section, and submit your "Accept" recommendation.

Reviewer #1: (No Response)

Reviewer #2: All comments have been addressed

Reviewer #3: (No Response)

2. Is the manuscript technically sound, and do the data support the conclusions?

Reviewer #1: (No Response)

Reviewer #2: Yes

Reviewer #3: (No Response)

3. Has the statistical analysis been performed appropriately and rigorously? 

Reviewer #1: (No Response)

Reviewer #2: Yes

Reviewer #3: (No Response)

4. Have the authors made all data underlying the findings in their manuscript fully available?

Reviewer #1: (No Response)

Reviewer #2: No

Reviewer #3: (No Response)

5. Is the manuscript presented in an intelligible fashion and written in standard English?

Reviewer #1: (No Response)

Reviewer #2: Yes

Reviewer #3: (No Response)

6. Review Comments to the Author

Reviewer #1: Fine Fine Fine Fine Fine Fine Fine Fine Fine Fine Fine Fine Fine Fine Fine Fine Fine Fine Fine Fine Fine Fine Fine Fine

Reviewer #2: (No Response)

Reviewer #3: The author have addressed all of my concerns. I have no more comments. I think the paper will benifit the scientific community.

7. PLOS authors have the option to publish the peer review history of their article (what does this mean?). If published, this will include your full peer review and any attached files.

Reviewer #1: No

Reviewer #2: No

Reviewer #3: No
